# Bone Disease in Multiple Myeloma: Biologic and Clinical Implications

**DOI:** 10.3390/cells11152308

**Published:** 2022-07-27

**Authors:** Zachary S. Bernstein, E. Bridget Kim, Noopur Raje

**Affiliations:** 1Center for Multiple Myeloma, Massachusetts General Hospital Cancer Center, Boston, MA 02114, USA; zbernstein@mgh.harvard.edu; 2Department of Pharmacy, Massachusetts General Hospital, Boston, MA 02114, USA; ebkim@partners.org; 3Harvard Medical School, Boston, MA 02115, USA

**Keywords:** multiple myeloma, denosumab, bone metastasis, bisphosphonates: zoledronic acid

## Abstract

Multiple Myeloma (MM) is a hematologic malignancy characterized by the proliferation of monoclonal plasma cells localized within the bone marrow. Bone disease with associated osteolytic lesions is a hallmark of MM and develops in the majority of MM patients. Approximately half of patients with bone disease will experience skeletal-related events (SREs), such as spinal cord compression and pathologic fractures, which increase the risk of mortality by 20–40%. At the cellular level, bone disease results from a tumor-cell-driven imbalance between osteoclast bone resorption and osteoblast bone formation, thereby creating a favorable cellular environment for bone resorption. The use of osteoclast inhibitory therapies with bisphosphonates, such as zoledronic acid and the RANKL inhibitor denosumab, have been shown to delay and lower the risk of SREs, as well as the need for surgery or radiation therapy to treat severe bone complications. This review outlines our current understanding of the molecular underpinnings of bone disease, available therapeutic options, and highlights recent advances in the management of MM-related bone disease.

## 1. Introduction: MM Pathogenesis

Multiple Myeloma (MM) is a malignancy of clonal plasma cells (PCs) localized in the bone marrow (BM) that mostly affects patients over 65 years old [1]. There is no clear cause of MM development; however, genetic and microenvironmental abnormalities play a role in its pathogenesis [2]. MM is the second-most-common hematological malignancy, accounting for 10% of hematological neoplastic disorders, with around 35,000 new cases and 12,000 deaths each year [3,4,5]. The proliferation of malignant monoclonal PCs in the bone marrow produces elevated levels of a serum, patient-specific immunoglobulin and its light chain (M-protein) [6,7].

MM arises from a premalignant asymptomatic condition known as monoclonal gammopathy of undetermined significance (MGUS). MGUS is defined as the accumulation of <10% clonal BM PCs, <30 g/L serum M-protein, and the absence of end-organ damage, which can include hypercalcemia, renal insufficiency, anemia, or bone-disease-related osteolytic lesions (CRAB) [8]. MGUS diagnosis has increased in incidence over the past 20 years, partially attributed to advancing detection methods [9,10,11]. MGUS can remain stagnant or progress into a more advanced, asymptomatic condition, recognized as smoldering or asymptomatic MM (SMM) [9,10]. SMM refers to the accumulation of >10% clonal BM PCs and/or >30 g/L serum M-protein without any CRAB symptoms. Until 2014, the diagnostic criteria of MM required one or more CRAB pathologies to occur in addition to the criteria for SMM to be considered a symptomatic and treatable disease [3]. In 2014, the International Myeloma Working Group (IMWG) updated their criteria for active and symptomatic MM to include patients with either >60% PC infiltration in the BM, >100 mg/L light-chain involvement, a >100 kappa/lambda light-chain ratio, or more than one focal bone lesion site, with or without the presence of CRAB symptoms [12].

In the last 20 years, treatment options for MM patients, such as immunomodulatory drugs (IMiDs), proteasome inhibitors, monoclonal antibodies, and autologous stem cell transplants, have paved the way for extending patient overall survival [13]. In 1997, the expected overall survival for MM patients was 2.5 years, whereas a 2013 study found that number to have increased to 5.2 years [14,15]. Although major advances continue in the field of MM research, the disease remains not only incurable in most patients, but the additional physical impact of bone disease poses daily threats to patient safety and wellbeing, even while in remission [16,17]. This review focuses on the cellular and molecular mechanisms of MM bone disease and provides an overview of the current treatment modalities.

SRE Definition, Cord Compression, and Fractures

Uncontrolled malignancies are the cause of most cancer-related mortalities [18]. Bone, specifically in the spine and ribs, is a common site for metastasis in breast, prostate, lung, kidney, and hematological cancers, which together account for around 350,000 deaths per year [18]. Due to increased vascularity, abundant growth factor, and prostaglandin production, the bone marrow microenvironment is an attractive space for the colonization of tumor cells [18]. In MM specifically, bone-disease-related lesions are the second-most-prevalent CRAB feature [19]. Over 80% of newly diagnosed MM patients will develop detectable bone disease [2,20]. 

The potential to improve MM disease detection and staging has been at the forefront of research for the last two decades. As recently as 2003, the IMWG recommended conventional skeletal survey as its method for staging and bone disease evaluation [21,22]. However, detection methods have since improved and evidence of a lack of its sensitivity has gained ground. Studies have found that to detect changes in bone mass via conventional X-rays, 30–50% of the bone needs to be destroyed [21,23]. Underestimation of bone destruction is a major problem that can lead to incorrect staging and delayed initiation of treatment [24]. Radiological-based skeletal survey accuracy is impaired by the surrounding soft tissue around the bone. Imprecision in detection has given way to other methods, such as MRI, fluorodeoxyglucose PET (FDG-PET), and PET-CT imaging, as more accurate modalities, although more expensive [21,25]. More sensitive testing, such as MRI and PET-CT imaging, shows appreciable bone disease involvement of 95% and 91%, respectively [26,27]. Due to the high-contrast material of the bone and bone marrow fat, low-dose CT provides better osteolytic lesion detection [28]. A multicenter study confirmed previous results that between 20 and 25% of patients with negative skeletal survey scans will show bone lesions using CT [21,28,29,30,31]. Although CT scans are the current standard-of-care procedure, it is important to acknowledge that these modalities may not be available everywhere.

The median overall survival of MM patients is 6 to 7 years [32]. One study found that survival is reduced to 2 to 3 years after the initial diagnosis of bone metastasis [33,34]. Not only do bone-disease-related osteolytic lesions result in a severe decrease in quality of life, higher costs of healthcare, and a decrease in functional independence, but it puts patients at risk of a life-threatening skeletal-related event (SRE), such as a pathologic fracture and disability, spinal cord compression, severe bone pain, and the need for surgical intervention [9,26]. Due to the proximity of the central nervous system to weakening and lytic bones, symptoms, such as paresthesia and burning sensations, are common among patients who have MM with bone involvement [35].

## 2. Myeloma Bone Disease

Bone tissue is made up of both organic and inorganic components [36]. The organic components include osteocytes, bone-lining cells, osteoclast cells (OCs), osteoblast cells (OBs), collagen fibers, proteoglycans, and glycoproteins [36]. The inorganic material consists of mostly calcium and phosphate and makes up 60% of bone mass [36]. Normal maintenance and remodeling of bone tissue in healthy humans is a continuous process and is finely balanced between the interplay of the OC and OB activity [26]. Maintaining a strong and light mineralized bone structure is a heavily regulated process within the body and, when disrupted, the marrow microenvironment is a common site for disease development [35].

OBs originate from mesenchymal stromal stem cells within the bone marrow and are responsible for the formation of structural bone material [36]. Located in the periosteum and endosteum, OBs produce the components of the extracellular matrix, such as structural macromolecules, including type-I collagen, proteoglycans, and cell-attachment proteins; OBs lay the foundation for bone mineralization [36]. Surrounding matrix vesicles deliver calcium, phosphate, alkaline phosphatase, adenosine triphosphatase, inorganic pyrophosphatases, among other proteinases, to catalyze hydroxyapatite crystal formation [36,37]. In the process, structural proteins, such as type-I collagen, act to guide and orient mineralization to maintain the shape and structure of all bones [36].

### 2.1. Basic Biology of Non-Diseased State (Expanding Pathogenesis)

Bone formation requires transcription factor Runx2/Cbfa1 activity and is responsible for mesenchymal stem cell differentiation into OBs [38]. Osterix, another transcription factor, also induces bone formation [38,39]. One study found that double-knockout (Runx2^−^/Cbfa1^−^) mice were deficient in OBs and bone formation [38,40]. However, the overexpression of Runx2 can stunt bone formation due to heavy regulation in the BM microenvironment [41]. Markers, such as collagen−1, are expressed due to the activation of Runx2/Cbfa1 during osteoblastogenesis and, therefore, can be clinically followed to track a patient’s disease [38].

OCs are large, multinucleated, specialized bone resorbing cells vital for normal bone remodeling [42]. They originate from hematopoietic precursors in the monocyte and macrophage lineage [42,43]. OCs are located within the Haversian canals, attached to the endosteal surfaces and calcium hydroxyapatite matrices, where they recede bone structure through proteolytic degradation and acid decalcification [42,43]. OCs accomplish these functions with proteins, including carbonic anhydrase II, calcitonin receptors, tartrate-resistant acid phosphatase, and lysosomal proteases [44].

The BM microenvironment is a well-endowed source of cytokines, such as interleukins, tumor necrosis factors (TNF), colony-stimulating factors, and growth factors that play key roles in structural regulation [36]. The understanding of the relationships between receptor activator of nuclear factor κB (RANK), its ligand (RANKL), and osteoprotegerin (OPG) have especially led to major advancements in bone homeostasis. RANKL is a cytokine and TNF family protein expressed by OBs; it binds to the RANK receptor on both precursor and mature OCs [43]. This interaction is necessary to initiate differentiation, activation, and survival of OCs [45]. Of note, it was found that rats devoid of RANK and RANKL genes had no OCs [45,46]. Therefore, the existence of OCs is dependent on both RANK and RANKL genes. OPG is another member of the TNF family expressed by OBs [2]. It functions as a protagonist and ‘decoy’ receptor to RANK by also binding RANKL as a regulatory step to minimize bone resorption and osteopenia [2]. In OPG knockout mice, early onset osteoporosis and vascular calcification have been shown to develop, confirming OPG’s role in maintaining bone integrity [2,47]. By inhibiting the RANK/RANKL interaction, OPG plays an important role in preventing OC-induced bone resorption [45]. Not only does it slow the proliferation of mature OCs, but it directly inhibits the differentiation of precursor OCs too [43]. The activation of both the macrophage colony-stimulating factor (M-CSF) and immunoreceptor tyrosine-based activation motif (ITAM) signaling pathways also play major roles alongside RANK and RANKL in OC activation [42,43,48,49]. M-CSF and ITAM, among other cytokines, act in tandem to regulate OC precursor differentiation and proliferation [43].

### 2.2. MM Diseased State

OC-driven bone destruction is not unique to cancer. Bone disease is common in many different conditions, including osteoporosis, which also results from the decoupling of OC and OB regulation [50]. In MM, malignant PCs also disrupt this careful balance that favors a net resorption of bone due to the widespread activation of OCs [26,51]. In fact, this disruption has been seen as early as in MGUS. It has been observed that patients with MGUS have higher rates of resorption and osteoporosis, measured by bone mineral density [52]. Conversely, evidence has shown that osteoporosis can increase the risk of MGUS [53,54,55].

In MM, tumor PCs produce both OC-activating and OB-inhibiting factors [56]. In fact, even in deep remission, OB activity is suppressed in MM [17]. MM differs from other malignancies with bone involvement because it exhibits little to no bone formation, whereas in metastatic breast and prostate cancers, both OCs and OBs are upregulated, emphasizing the need to protect and rebuild the structural integrity of bones in MM [50,57].

The complete mechanisms by which MM cells inhibit OB formation and differentiation remain unclear. However, many signaling factors have been identified as bone formation antagonists, such as the Wnt signaling pathway inhibitor Dickkopf1 (DKK1) and interleukin-7 (IL−7) [58]. DKK1 is secreted directly from tumor PCs to inhibit OB differentiation [59]. Increased expression of DKK1 has been shown to be correlated with bone-disease-related osteolytic lesion severity, whereas IL−7 secreted by BM stromal cells suppresses Runx2/Cfba1 promoter activity, a necessary transcription factor for OB formation [44,58,59,60].

Runx2/Cfba1-mediated transcription and osteoblastogenesis have been shown to be inhibited by MM cells in vitro [58]. As a result, OB precursor formation and differentiation were reduced, as seen by lower expression of alkaline phosphatase, osteocalcin, and collagen-I [38,58]. This finding was seen in mature OBs as well [61,62]. Results showed that MM cells can directly inhibit OB proliferation and even make OBs more prone to apoptosis [61,62]. Runx2/Cfba1 also stimulates OPG secretion by OBs [58]. Therefore, the inhibition of Runx2/Cfba1 not only reduces the development and expansion of OBs, but it allows OC expansion to be even further unregulated. Runx2/Cfba1 activity can also be inhibited via cell-to-cell contact between osteoblastic progenitor cell membrane protein VCAM-1 and MM cell’s VLA-4 protein, which have been seen to upregulate RANK following contact [38,58].

Tumor-PC-related OC activation pathways are also well documented. MM BM microenvironments are known for their ability to foster favorable conditions to perpetuate uncontrolled tumor growth through processes, such as increased angiogenesis and increased circulation of growth factors, such as IL−6 and vascular endothelial growth factor (VEGF) [2,63]. More specifically, MM PCs produce cytokines specifically to develop a microenvironment that promotes malignant proliferation and cell survival (Figure 1) [38,51,64,65]. For example, RANK has been shown to increase vascular permeability and angiogenesis through activation of nitric oxide synthase in endothelial cells to improve the circulation of tumor-promoting factors [66]. The release of humoral factors, such as IL−6, among other cytokines stimulates the upregulation of RANKL, thus, promoting OC activation, bone resorption, and the release of bone calcium and growth factors stored in the bone matrix (Figure 1) [67]. Other factors, such as IL−1, IL−3, E series prostaglandins, and TNF-α, increase remodeling and resorptive activity resulting in a vicious cycle of bone turnover, more growth factor dissemination, and hypercalcemia [68]. Found in one study, tumor PCs can also stimulate RANKL upregulation while downregulating OPG expression, which has even more severe side effects than in healthy counterparts and asymptomatic MM patients, (Figure 1) [51,69]. Levels of serum OPG usually indirectly correlate with MM bone disease severity [51,70]. Similarly, one group discovered that RANKL/OPG ratios are also indirectly related to expected survival, showing unfavorable implications for MM patients [2,71].

Data suggest that the anabolic effect of anti-MM therapy has been shown to mitigate aggressive bone resorption. Proteosome inhibitors (PIs), such as bortezomib, have been observed to do so through stimulating Runx2/Cbfa1 activity [72,73]. Bortezomib’s anti-MM activity further allows a recovery of OB levels, seen through an increase in alkaline phosphatase expression [72,73,74,75]. Immunomodulatory drugs (IMiDs), such as lenalidomide and pomalidomide, in addition to proteosome inhibitors, have been shown to reduce RANKL production by blocking its production [76,77]. IMiDs specifically inhibit OC formation and RANKL upregulation [78].

Hypercalcemia of malignancy (HCM) is a common complication of various advanced cancers, including MM, lung, breast, and kidney cancers [79]. While it can present as mild to life threatening, HCM is often related to a more advanced and aggressive disease burden, especially in MM [79,80]. Prognostically, HCM is associated with significantly inferior survival rates (26 months vs. 48 months, *p* < 0.001), although other factors, such as high-risk FISH cytogenetics (del17p, t [4; 14], t [14; 16]), can influence survival and ISS disease staging [80]. Approximately one-third of patients will experience such metabolic complications, resulting from extensive bone destruction directed by the MM cell secretion of RANKL and other pro-destruction cytokines, such as DKK1, tumor necrosis factors, and macrophage inflammatory protein (MIP-1α) [79,80]. All of these cytokines can also be over-expressed by other cells surrounding the tumor microenvironment, leading to an increased accumulation of calcium in the serum, resulting in life-threatening complications, such as dehydration, confusion, acute renal insufficiency, or even a comatose state [35].

### 2.3. Available Treatment Agents

OC-specific targeting agents have been a focus of treatments for bone-related diseases, such as osteoporosis, among numerous other malignancies, including breast cancer, prostate cancer, and MM, for many decades. Bisphosphonates (BPs) were the first drug class to both treat and prevent bone-disease-related osteolytic lesions. Randomized placebo controlled trials revealed that BPs show efficacy in treating and preventing SREs [26]. Other agents are in development to target upregulated pro-osteoclastogenic and/or anti-osteoblastogenic molecules. In MM, such targets include: IL−3, activin A, TRAF6, and BTK [76].

Molecularly, BPs are related to inorganic pyrophosphates and are recognized by their phosphorus–carbon–phosphorus structural backbone [81]. In contrast to inorganic pyrophosphates, BPs are synthetic and hydrolysis-resistant molecules that have a high affinity to calcium and, therefore, target areas of high resorption on bone hydroxyapatite surfaces [82,83]. Mechanistically, BPs restrict OC activity via the mevalonate pathway of protein prenylation through inhibiting farnesyl pyrophosphate synthase, thereby inducing OC apoptosis and preventing bone resorption [84]. Significant advancements in the MM and BP fields, following the development of higher-potency nitrogen-containing agents, such as intravenous pamidronate and zoledronic acid, have proved momentous in improving patient bone pain relief, SRE prevention, and hypercalcemia, among other aspects of patients’ quality of life [81,85].

Pamidronate was the first BP to show a clinical benefit in MM [81]. Before pamidronate, another BP, clodronate, showed efficacy in delaying osteolytic lesion development, but did not significantly improve bone pain or the incidence of bone fracture [86]. In a randomized study, patients with stage III MM and at least one osteolytic lesion were given pamidronate every four weeks for nine cycles [85]. It was found that the time to the first SRE, pathologic fracture, and radiation treatment to a lesion were all significantly less than in the placebo group (*p* = 0.001, *p* = 0.006, *p* = 0.05) [85]. These findings led to its Food and Drug Administration (FDA) approval in 1995 [85]. Over the course of the study, the placebo group experienced more SREs, more frequent hypercalcemia, and a worse quality of life [85]. Even within the first month of treatment, bone pain and analgesic-drug use was reduced in the pamidronate group [85], although overall survival was no different [85]. Similar results were found in another randomized, double-blind study observing pamidronate efficacy over 21 cycles [87]. In this study, pamidronate reduced overall SRE incidence after 12, 15, 18, and 21 months versus the placebo group in MM [87]. Pamidronate considerably improved the quality of life and prognosis of MM patients, especially those with heavy bone disease involvement; however, its two- or four-hour intravenous (IV) administration limits accessibility [88].

Zoledronic acid (ZA) is a more potent BP that has also shown efficacy in clinical trials [81]. ZA became the first BP to show efficacy in solid tumors (excluding breast cancer), such as prostate and non-small-cell lung (NSCLS) cancers [88,89,90]. The median time to progression improved but was not significantly different in the ZA treatment group over the pamidronate treatment group, 136 days vs. 113 days [91]. Similar non-significant results were found in another study, including the mean number of annual SRE incidents in ZA versus pamidronate, 1.00 SREs vs. 1.39 SREs [92]. Overall, SRE prevention was similar between the two BPs; however, in patients with moderate to severe HCM, ZA showed significant benefit over pamidronate in mediating calcium levels [92]. ZA has also shown evidence of anti-MM activity. In comparison to clodronate, the MRC Myeloma IX trial showed that ZA therapy reduced mortality by 16% and improved overall survival to 50 months vs. 44.5 in the clodronate group (*p* = 0.04) [93], meaning that both BP and anti-MM therapies can positively impact patient survival and SRE prevention. SRE incidence decreased with ZA, 27% vs. 35% (*p* = 0.0004) [93]. The findings of this trial prompted the National Comprehensive Cancer Network (NCCN) and the International Myeloma Working Group (IMWG) to recommend BPs to treat symptomatic MM along with anti-MM therapy [81,94,95]. The other advantage of ZA is its considerably shorter safe administration time of 15–30 min [88]. Based on these findings, ZA received FDA approval in 2002 and was incorporated into American Society of Clinical Oncology guidelines for MM bone disease following two randomized studies that demonstrated noninferiority to pamidronate in number and time to development of SREs [85,88,96].

Measurement of bone turnover markers, such as urinary or serum n-telopeptide of type-I collagen (uNTX or sNTX, respectively) can help inform treatment decisions and predict patient disease status, risk profile, bone metabolism status, and BP activity [97]. High baseline sNTX levels are generally associated with elevated risk of SRE [98,99] One study found that in patients who achieved a partial or complete response from anti-MM therapy and whose baseline sNTX levels were also suppressed, sNTX levels remained suppressed [100]. Another study that tracked how uNTX levels related to overall survival and SRE risk between patients receiving either ZA, pamidronate, or placebo found that when uNTX levels normalized (defined as <64 nmol/mmol creatinine) in patients treated with ZA, patient’s risk profile decreased [101]. Those whose uNTX levels normalized were found to have a decreased risk of death by 48% in breast cancer, 59% in HRPC, and 57% in NSCLC (*p* = 0.0017, *p* < 0.0001, *p* = 0.0116, respectively) [101]. This information confirms a 50% reduction in the risk of experiencing an SRE in breast cancer found in another study (*p* = 0.0031) [33]. Even after the cessation of BP therapy, those who achieved a clinical response to anti-MM therapy continued to have suppressed sNTX levels 6 months following their last ZA dose [100]. To determine better ZA-dosing schedules for the spectrum of patient disease risks and bone metabolism, the Z-MARK study stratified patients who received BP therapy for between 1 and 2 years based on baseline uNTX levels [98]. In this study, ZA 4 mg was given either every 4 weeks (patients with uNTX ≥ 50 nmol/mmol creatinine) or every 12 weeks (patients with uNTX < 50 nmol/mmol creatinine). Patients initially treated every 12 weeks were switched to treatment every 4 weeks if one of three events occurred, including: disease progression, SRE incident, or uNTX level increase [98]. In those cases, a more aggressive, monthly ZA infusion may be needed to treat bone resorption [98,102]. Overall, 32.5% of patients (38 of 117 patients) were switched to a more frequent ZA treatment schedule, while the rest (79 patients) remained in the ZA 12-week treatment group [98]. This study found that uNTX levels were not necessarily predictive of SRE incidences, even in patients with elevated uNTX levels, but that bone metabolism can be maintained with ZA 4 mg administration every 12 weeks [98]. Similarly, another study assessing the risk profile of SREs between patients treated with ZA either every 4 or 12 weeks found no significant difference in MM patients (*p* = 0.14) [103], therefore, confirming the safety and efficacy of the less frequent treatment interval.

#### Bisphosphonate Safety Precautions

Patients receiving long-term BP therapy are at risk of osteonecrosis of the jaw (ONJ). ONJ is a serious and painful adverse event recognized as exposed and necrotic bone in the maxillofacial region that persists for eight weeks related to BP therapy [104,105]. The majority of cases take around four months to heal [104,105]. Severe complications in light of invasive dental procedures can occur as the molecules remain on maxillofacial surfaces, although the mechanism of action is not yet fully understood [4]. Preventative strategies, such as avoiding tooth extractions (exclusive of root canals, dental cleanings), while on BP therapy and/or holding BP therapy for 90 days before and after invasive elective procedures has shown effectiveness in reducing ONJ incidence and is, thereby, recommended by the IMWG [106]. Studies suggest that between 4 and 11% of patients will develop ONJ and the risk of development increases directly with prolonged BP exposure [4,107]. Pamidronate 30 mg also showed reduced incidence of ONJ compared to pamidronate 90 mg [108]. Moreover, there is little difference in incidence rate between ZA and other BPs [109].

Renal toxicity is another limiting factor for patients undergoing BP therapy as it can induce acute renal damage [110,111]. BPs are excreted unmetabolized through the urine and can, therefore, form insoluble precipitates with calcium in the renal tubuli and, most frequently, result in acute tubular necrosis [81,91]. Therefore, previous hypercalcemia-related renal impairment can exacerbate dysfunction. The accumulation of BP molecules in the renal tract facilitates renal dysfunction and relates directly with the length of BP treatment [81]. For that reason, the FDA recommended BP dosage adjustment based on creatinine clearance [112]. In a key phase III study, ZA showed elevated renal impairment in the 8 mg dosage group compared to pamidronate (90 mg in 250 mL for 2 h), suggesting that a dose reduction to 4 mg in 100 mL over 15 min was safer than IV infusion of ZA in 5 min [91].

BPs serve to prevent bone loss but can adversely affect bone formation and quality, so the development of agents with the ability to improve both functions is currently an unmet need [113]. Alternative agents, such as the sclerostin (Scl) inhibitor and monoclonal antibody, romosozumab, have demonstrated both anticatabolic and anabolic-promoting effects in treating osteoporosis [113]. Sclerostin functions to antagonize the Wnt/β-catenin pathway to inhibit bone formation, as well as upregulate RANKL levels [113,114,115]. Data suggest that the activation of both the Wnt and β-catenin pathways, via a Scl inhibitor, promote osteoblast formation and survival and, thus, are a promising therapeutic target for combatting MM bone disease [113]. Pre-clinical studies in MM have shown that the deletion of the gene that encodes sclerostin, SOST, prevents MM bone disease in immune-deficient mice [113]. Activin A is another potential target to treat bone disease. Sotatercept, a recombinant activin receptor that binds activin A and GDF11 with high affinity, is currently being investigated in early trials [116]. Activin A is also involved in OC activation and OB inhibition and is upregulated in MM [113,117]. Sotatercept has been shown to increase bone mineral density in phase-1 studies in MM [116]. Although these agents show promising data in preclinical and early clinical studies, BPs remain the mainstay treatment of bone disease.

## 3. Denosumab

Denosumab is a more recently developed agent for treating metastatic bone disease. Alternatively to BPs, denosumab is a fully human monoclonal antibody administered subcutaneously that targets RANKL to inhibit osteoclastogenesis and OC-mediated bone resorption [26]. Unlike ZA, pamidronate, clodronate, and other BPs, denosumab does not persist in bone tissue nor does it rely on renal clearance, rather, reticuloendothelial clearance [26,118]. It also has a circulatory half-life of 26 days, similar to other monoclonal antibodies [26,118]. Despite efficacy shown by ZA in prolonging SREs and reducing skeletal complications, bone metastasis still occurs [91]. There are also limitations to ZA, such as renal complications and its IV administration, that pose problems for patients, suggesting the need for therapies that further minimize toxicities and provide more easily administered cocktails for treating metastatic bone diseases. Denosumab was first established as effective in the treatment of osteoporosis. A Phase III study, FREEDOM, enrolled 7886 women with osteoporosis where denosumab 60 mg was given every six months and demonstrated superiority to placebo in its efficacy, reducing vertebral, nonvertebral, and hip fracture risk [119]. In 2010, the FDA approved denosumab for the treatment of postmenopausal women with osteoporosis at high risk for fracture as a result of this trial.

### 3.1. Denosumab in Metastatic Bone Disease

Denosumab’s effectiveness in preventing SREs in metastatic bone disease was tested in three separate, but identically designed, Phase III randomized, double-blind trials in breast cancer, prostate cancer, and solid tumors (excluding breast and prostate) or MM (Table 1). Patients were treated with either denosumab 120 mg s.c. or ZA 4 mg i.v. every four weeks along with recommended calcium and vitamin D supplements if necessary [120,121,122]. The endpoints of the studies were noninferiority and superiority tests to evaluate the time to first SRE. The third study in solid tumors plus MM and excluding prostate and breast cancers (mainly lung and MM) enrolled 1776 patients and found similar results as the others in the time to the first SRE being longer with denosumab amongst all patients (20.6 months vs. 16.3 months; 0.84 with 95% CI: 0.71–0.98; *p* = 0.0007 noninferiority; *p* = 0.06 superiority) [120]. By tumor stratification, both patients with MM and non-small-cell lung cancer showed equivalence in delaying the first SRE, not superiority [120].

In all three studies, denosumab demonstrated a reduction in renal toxicity but no difference in overall survival compared to ZA [120,121,122]. In patients with a baseline creatinine clearance ≤60 mL/min, ZA showed a noticeably higher risk of renal AEs (including acute renal failure, increased creatinine, and increased urea ect.), 21.6% of patients in the ZA arm versus 11.3% in the denosumab arm [120]. Due to the related renal toxicity in ZA, 8.9% of doses were withheld because of elevated serum creatinine [120]. One meta-analysis of all three studies of denosumab versus ZA in metastatic cancer also found a similarly appreciated risk of renal AEs in the ZA treatment arm compared to denosumab, 11.8% vs. 9.2% (*p* = 0.002) [124]. Despite dose reductions in ZA, potentially renal-toxicity-related AEs and flu-like syndrome were both higher than that of denosumab [120,121,122]. Incidences of creatinine clearance reduction were also higher in the ZA arm [122]. However, incidences of hypocalcemia, a recognized AE, occurred more frequently in the patients treated with denosumab [120,121,122]. Of note, most cases were asymptomatic and only a small number of participants required a calcium supplement [120,121,122]. Denosumab’s lack of renal clearance and no evidence of renal effects present it as a safer therapeutic option for patients with renal impairment [120].

### 3.2. Denosumab in MM

The third phase III study comparing the efficacy of ZA and denosumab in metastatic cancer, excluding breast and prostate cancer, found there to be no difference in overall survival in the total patient cohort; however, in MM specifically, overall survival was found to be worse in denosumab-treated patients [120]. Although equivalence was shown between ZA and denosumab in MM in delaying the first SRE, MM was not given indication for denosumab [26,120]. These results may be due to important limitations of the study. The MM patient population was a small cohort limited to 10% of the broader study while lung cancer patients counted for 40% [26,120]. Further, due to an imbalance in MM patient prognostic factors, anti-MM therapies, and withdrawals, it is challenging to make conclusions about the efficacy of SRE prevention and treatment of denosumab versus ZA in MM [120,123].

An international, randomized, and double-blind study only enrolling MM patients was conducted to address these limitations. This phase III study enrolled 1718 newly diagnosed patients and stratified by prognostic factors, intent to undergo autologous stem cell transplant (ASCT) and anti-MM therapy in a control and double-blind analysis of denosumab versus ZA [123]. Among other inclusion criteria, patients who presented with at least one x-ray or CT-confirmed bone lesion and a creatinine clearance ≥30 mL/min (due to ZA-dosing restrictions) were included [123]. As with the previous three trials, the primary and second endpoints were whether denosumab is noninferior and superior, respectively, to ZA in time to the first SRE (defined as a pathologic fracture) [123]. Likewise, dosing levels and treatment schedules were consistent to the previous three phase III studies (s.c. denosumab 120 mg and i.v. placebo every four weeks or s.c. placebo and intravenous ZA 4 mg) [123]. The first endpoint was met, as denosumab was found to be noninferior to ZA in delaying the time to the first SRE (22.83 months vs. 23.98 months; HR = 0.98 (0.84–1.14); *p* = 0.01 noninferiority) [123]. Overall survival, an exploratory secondary endpoint, was also found to be similar between treatment groups [123]. However, progression-free survival favored denosumab-treated patients 46.1 months vs. 35.4 months (*p* = 0.036), presenting denosumab as an effective and safe treatment in MM [123]. Renal-toxicity-associated AEs was more common in the ZA arm (17% vs. 10%; *p* < 0.001) and even further pronounced in patients with baseline renal impairment (baseline creatinine clearance ≤ 60 mL/min), 26% in the ZA arm versus 13% in the denosumab arm [123]. Hypocalcemia was elevated with denosumab treatment, 17% vs. 12%, but most cases were grade 1–2 [123], whereas ONJ occurred more often with denosumab, although not significantly, 4% vs. 3% [123]. A higher risk of ONJ with denosumab was confirmed in a study in osteoporosis patients (HR: 3.49, 95% CI 1.16 to 10.47, *p* = 0.026) [125].

These findings are in support of denosumab as a standard-of-care therapeutic agent for MM. Although the findings were limited by a creatinine clearance of ≥30 mg/min due to the study being blinded, denosumab is a compelling option for patients with renal insufficiency, thus, establishing denosumab along with bisphosphates as integral as adjuvant therapies.

Further exploratory studies have confirmed denosumab’s efficacy, not only in SRE prevention, but in measuring progression-free survival (PFS). Extending from the MM only denosumab versus ZA phase III double-bind study, a significant improvement in PFS in denosumab versus ZA was concluded in newly diagnosed MM patients [126]. These results were especially significant in patients with intent to undergo ASCT with denosumab compared to ZA (46.1 months vs. 35.7 months; *p* = 0.002) [126]. Of those who had frontline triplet induction therapy or bortezomib-only therapy (without IMiD) saw the biggest PFS benefit [126]. A potential synergistic effect between denosumab and bortezomib was suggested by the fact that both interact with RANKL and bortezomib is known to reduce DKK1 levels [126,127,128]. Proteosome inhibitors (PIs), including bortezomib, have shown previous evidence for suppressing osteoclast differentiation and stimulating osteoblastogenesis to promote bone formation [77,129,130,131]. Of note, patients with ASCT intent are generally younger and can endure more intensive anti-MM treatment. To account for older patients, it was found that those <70 years of age still experienced a PFS benefit with denosumab; however, this information was not statistically powered [126]. Although this meaningful clinical benefit was seen in newly diagnosed patients given denosumab therapy, there remain many unanswered questions and endpoints that require further investigation.

## 4. Guideline Recommendations

Bisphosphonates have served as the standard of care for the prevention and treatment of mm bone disease. However, a more robust understanding of the pathophysiology of antiresorptive therapies, such as RANKL inhibitor denosumab, has led to updated treatment recommendations [111]. The 2021 IMWG guidelines recommend BP therapy, specifically ZA or pamidronate, in all patients with active MM, with regular kidney monitoring, whether or not MBD is present [111]. Pamidronate is also recommended for patients with severe renal impairment, including those on dialysis [132]. Clodronate is not approved in the United States because ZA was proven to be superior for preventing SREs and improving survival [12,93]. It is approved for use in Canada and the United Kingdom, however [12,93]. The ASCO guidelines also recommend BPs after the MRC IX Trial showed that BP therapy showed benefit in patients without lytic lesions [12,93,133]. In patients with MGUS or SMM, BP treatment is only recommended if a patient also has osteoporosis, unless they have high-risk SMM or SMM with an MRI or PET/CT-confirmed lesion [111]. Then, patients can be considered for BP therapy similar in schedule and dosing to that of active MM [111]. Patients treated with BPs are also recommended to receive calcium and vitamin D supplementation, once calcium levels have been normalized [111]. Denosumab is recommended for newly diagnosed MM, and relapsed and refractory MM with bone disease involvement [111]. In patients with renal impairment and HCM, denosumab is recommended [92,111]. Even for patients with creatinine clearance <30 mL/min, denosumab can be given under close monitoring [111]. Patients with MGUS or solitary plasmacytoma who also have osteoporosis are recommended to receive denosumab for bone disease treatment and prevention [111].

Regarding dosing and treatment schedules, it is recommended that ZA 4 mg is administered intravenously every 3–4 weeks for 15 min for those with symptomatic MM for SRE prevention and treatment [111]. It is also recommended that pamidronate 30 mg or 90 mg is administered every 3–4 weeks as well for the same indication [111]. In patients with HCM, ZA is recommended over pamidronate due to proven superiority [134]. However, in MM with HCM that is refractory to BPs, denosumab is recommended [111]. ZA is also favored over pamidronate due to its shorter infusion time. Treatment with ZA is recommended to continue for 12 months. If a very good partial response or better is achieved, then the frequency of dosing can be reduced to every 3, 6, 12 months, or until discontinuation or the physician’s discretion [111]. After discontinuation, BP treatment should be reinitiated at the time of biochemical relapse to lower the risk of an SRE [111]. Regarding ONJ, it is recommended that a comprehensive dental exam is conducted before BP therapy and current oral infections must be healed [12]. While on therapy, invasive dental exams should be avoided. If a procedure is unavoidable, denosumab should be discontinued 30 days before and held until the incision is fully healed [111]. Due to subjective risk factors, changes to treatment should be considered on a patient-by-patient basis. Subcutaneous administration of denosumab 120 mg once a month with calcium and vitamin D supplementation is recommended as well [111]. The duration of treatment can extend until unacceptable toxicity occurs [111]. Changes to dosing schedules are only recommended after 24 months of treatment on a patient-by-patient basis [111]. At this point, it is also recommended that discontinuation of denosumab is avoided due to data showing a rebound of elevated resorptive activity, characterized by osteoclastogenesis, rapid decline in bone mineral density, and a greater risk of vertebral fractures between 6 and 12 months after treatment extinction [111,135,136]. If denosumab is discontinued, the European Calcified Tissue Society recommends beginning BP therapy to minimize the magnitude of OC rebound [135]. This rebound effect is not yet completely understood, but it is suspected that either the dysregulation of Wnt inhibitors SOST and DKK1 are involved, or that a dormant pool of osteoclast precursors is stimulated following the suspension of denosumab, leading to a surge in RANKL levels [137]. One case-control study found higher bone turnover markers and lower SOST levels in denosumab cessation versus treatment-naïve patients [137,138]. Data to support the discontinuation of denosumab are limited but, based on previous and relevant clinical data from denosumab in osteoporosis, the IMWG recommends a single intravenous dose of ZA at least 6 months following discontinuation or another denosumab 120 mg injection at the same timepoint. More clinical information is needed.

Denosumab 120 mg costs nearly USD 2000 per dose in the United States, whereas ZA 4 mg and pamidronate 90 mg cost approximately USD 50 and USD 30, respectively [12]. Significant differences in price warrant consideration in choosing a treatment regimen. Factors, such as efficacy profile, treatment toxicity, economic viability, intravenous infusion chair accessibility, and infusion time, all need to be considered. Regarding cost effectiveness of ZA versus denosumab, one study set out to compare value propositions from the perspectives of patients, payers, and society [138]. Given real-world data and from similar studies analyzing the cost effectiveness of denosumab versus ZA in solid tumors, incremental quality-adjusted life-year and net monetary benefits from each treatment were quantified [138]. The study concluded that denosumab is a cost-effective treatment option given its benefits in SRE prevention and progression-free survival based on direct and indirect medical costs [123,138]. Given this information, there is support for denosumab’s value as a therapeutic option over ZA; however, projection-based analysis has its own limitations and, therefore, more observation is needed to provide a conclusive comparison. It is also important to acknowledge cost feasibility disparities in clinical settings that may restrict access to denosumab, making ZA a more attractive regimen in that case.

## Figures and Tables

**Figure 1 cells-11-02308-f001:**
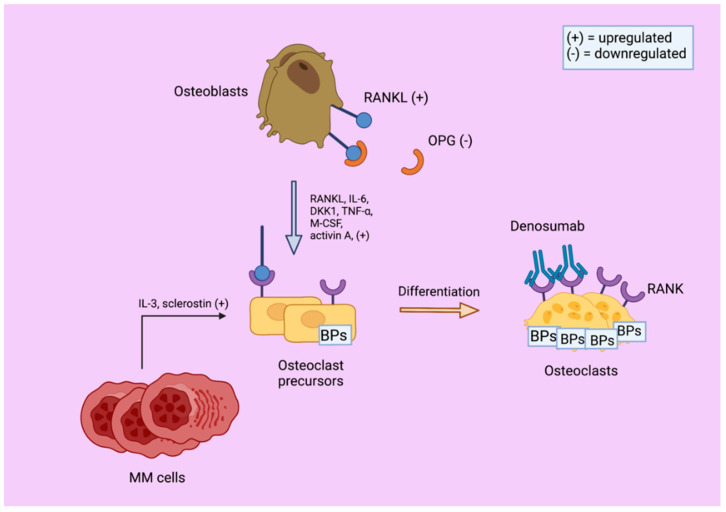
An overview of the interplay between multiple myeloma (MM) cells, osteoclasts, and osteoblasts in multiple myeloma, which favors bone resorption. MM cells induce IL−6 and DKK1 cytokine upregulation in osteoblasts, which, in effect, upregulates the osteoclast-activating protein, RANKL, while downregulating the anti-resorptive protein, osteoprotegerin (OPG). OPG binds and inhibits RANKL, while MM cells also agonize IL−3 and sclerostin productions. Denosumab directly inhibits RANKL while bisphosphonates (BPs) (i.e., clodronate, pamidronate, and zoledronic acid) inhibit osteoclasts and osteoclast precursors to prevent osteoclast-induced bone resorption. Created with BioRender.com (accessed on 7 May 2022).

**Table 1 cells-11-02308-t001:** Phase III randomized studies of denosumab versus zoledronic acid in solid and hematologic malignancies.

Treatment ^a^	Disease Group	*n*	Median Time to First SRE (Months)	Hazard Ratio	Renal Toxicity ^d^
Denosumab	Breast cancer [122]	1026	NR ^c^	0.82 (0.71−0.95); *p* < 0.001	4.9%
	Prostate cancer [121]	950	20.7	0.82 (0.71–0.95); *p* = 0.0002	16%
	Solid tumors ^b^ [120]	886	20.6	0.82 (0.71–0.98); *p* = 0.0007	8.3%
	Multiple Myeloma [123]	859	22.83	0.98 (0.85–1.14); *p* = 0.01	10%
Zoledronic acid	Breast cancer [122]	1020	26.4	−	8.5%
	Prostate cancer [121]	951	17.1	−	15%
	Solid tumors [120]	890	16.3	−	10.9%
	Multiple Myeloma [123]	859	23.98	−	17.1%

^a^ Treatment included subcutaneous denosumab 120 mg and intravenous placebo every four weeks or subcutaneous placebo and intravenous ZA 4 mg. ^b^ Tumors including lung and multiple myeloma and excluding breast and prostate. ^c^ Endpoint not reached (NR). ^d^ Incidence of renal adverse event and elevations in serum creatinine.

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
