# Peer review of "Bone Disease in Multiple Myeloma: Biologic and Clinical Implications"

_cells, 2022, doi:10.3390/cells11152308_

Round 1

Reviewer 1 Report

The manuscript is a review of the pathophysiology of bone disease in multiple myeloma, as well as treatment suggestions. In general the review is comprehensive, but lacks any discussion of links between MGUS and osteoporosis, which have been published. The definition of SMM, given at line 39 should state "and/or" as the sentence structure implies that both >10% PCs and 3g/dl M protein needs to be present.  Discussion regarding use of bone survey should be moved up to when imaging first discussed, and a little more depth as to the limitations, but also there should be acknowledgment that advanced imaging may not be available in many parts of the world. 

The discussion of bone biology leaves out bone lining cells and osteocytes, which should at least be mentioned in a publication that is aiming to be comprehensive.

More space is given to denosumab vs bisphosphonates, although the latter are significantly less expensive. No recs for calcium and VitD are given for BPs, but that is recommended and the trials leading to BP approval required that these be given.  No mention is given as well that pamidronate in particular can be given in patients with renal failure and even on dialysis. Also missing is data showing that ONJ may actually be much more common with denosumab DOI: 10.1002/jbmr.4472. Also, there should be more explanation as to why denosumab can lead to rebound bone loss quickly, when stopped, so that readers are aware of this issue. One area that is not mentioned, even to state that there is little data, is how to use treat recurrent bone disease, particularly in patients who have had previous years of either bisphosphonates or denosumab. Another omission from the discussion of dosing schedule that really stands out is no mention of the study by Himelstein et al using q 4 week vs 12 wk of zolendronic acid showing efficacy and safety of the longer interval. PMID: 28030702

The figure is not particularly helpful and could easily be omitted but if included should be revised to one that is easier to understand (there are a number that could be modelled, e.g. https://doi.org/10.1530/EJE-18-0056

Grammar is also an issue. Line 66, "Although" should be changed or just omitted. Line 93 needs an "and". Sentence line 95-96 doesn't make sense. Line 107 is redundant to line 109. Line 159-160 is not a sentence.  Line 185 "its" should be "theirs", line 281 the phrase "in patients" both starts and ends the sentence, so one usage should be removed.  Sentence 458 assume intent is to say "hypercalcemia that is refractory to bisphosphonates" but that is not clear.

However, with some editing and additions this will be an excellent review

Author Response

Reviewer 1 revisions:

  • Regarding the previous omission of the link between MGUS and osteoporosis, this has now been addressed between lines 175-178.
  • Line 39 now includes and/or instead of the previous comma in the definition of SMM.
  • Regarding the placement of information on scanning methods, I agree that their acknowledgment should occur in one paragraph rather than in two as previously written. On line 47, I deleted the initial reference and rather elaborated between lines 71-82 as I believe this information is better suited in the SRE definition section rather than in the introduction which is more of a general overview versus specificity to bone detection. Acknowledgement to the availability of more expensive and sensitive methods was now added to better portray accessibility as a constraint in some regions of the world (lines 81-82). I also elucidated about the limitations on skeletal survey sensitivity on lines 71-74 to provide more concrete evidence and an example to demonstrate its clinical limitations rather than just mechanical constraints of the physical machine.
  • On line 94, I added osteocytes and bone lining cells to provide a better and more comprehensive picture of all of the cells involved in bone homeostasis rather than limiting the discussion to osteoblasts and osteoclasts.
  • To the suggested edit of the space given to bisphosphonates versus denosumab, while I also agree with the amount of space given, I believe now that a broader picture of therapies is given with the anti-sclerostin antibody discussion among further discussion of bisphosphonates on lines 270-272 and 383-395 enhances the understanding of the field while offloading the emphasis of denosumab as the only other alternative to bisphosphonates. Also on lines 584-595, I add a discussion of the cost effectiveness of the therapies to outline further information to consider especially about prices.
  • On line 534, I added the recommendation of vitamin D and calcium supplements for those on bisphosphonate therapy as suggested to better inform readers about supportive regimens while on bisphosphonates.
  • On lines 525-526, I added information that pamidronate can be given for those on dialysis to better emphasize it’s efficacy for patients with concomitant health conditions.
  • Regarding the potential elevated risk of ONJ with denosumab, I added the suggested information on lines 493-494 as a way to provide a more thorough analysis of the potential downsides of denosumab.
  • On lines 567-574, I address more comprehensively why the denosumab bone loss rebound occurs through a pathway driven understanding which was lacking before.
  • On lines 551-552, I also comment on treatment strategies for those who have recurrent bone disease.
  • Regarding the Himelstein study, information was added between lines 353-356 to address the omission of the efficacy of different zometa treatment schedules on safety.
  • As for the figure, we agreed that simplicity would improve the understanding of the interactions between myeloma cells, osteoblasts and osteoclasts. Specifically, we chose to better depict the physical contact between cells while adding more cytokines relevant to disease development. Also, removing more of the arrows helped us provide a clearer picture of the bone marrow microenvironment in the diseased state.
  • Regarding the “although” on line 65, I removed it as to improve the sentence quality and flow.
  • We also added the “and” on line 96.
  • Between lines 106-108, I revised the sentence to make more logical sense and flow much smoother. This was previously on lines 95-96.
  • Regarding the redundancy of 121-123 (previously 107-109), we added a conjunction “and” to convey more information in one sentence rather than repeating in the following one.
  • In lines 175-176, (previously 159-160), we restructured the sentence to make more sense and flow better.
  • I lines 219-220 (previously 185), I changed the sentence and now “their” makes more grammatical sense.
  • In line 339 (previously 281) I removed the second “in patients”.
  • On line 584 (previously 548) I clarified the sentence to address HCM in patients refractory to bisphosphonates which was not evidence previously.

Thank you for your time in providing suggested edits to our paper! Overall, we think they all were helpful and relevant to our mission in writing this paper.

Reviewer 2 Report

The present review outlines the current understanding of the molecular underpinnings of bone disease and available therapeutic options. 

The review is well written and easy to follow. However, there are other agents targeting osteoclasts in clinical trial (see Bolzoni M et al, 2018, Expert Review of Hematology). Morever, the authors should at least mention the agents demonstrating the anabolic effects on bone (e.g proteasome inhibitors, anti sclerostin antibodies)

- the paper is well written and easy to follow. However, the topic is not new and the paper does not provide a significant contribution in the field.

- the authors mainly focused on agents targeting osteoclasts. There are other agents (e.g proteasome inhibitors, anti-sclerostin antibodies) that showed the anabolic effect on bone. These agents should be at least mentioned (short paragraph or table).

- there are other molecules expressed on osteoclasts that could be targeted by pharmacological agents. Some of them displayed good results in clinical and preclinical models (see Bolzoni M, Expert Review of Hematology, 2018 PMID: 29495905).

Author Response

Reviewer 2 revisions:

Thank you for the suggested edits and providing a perspective that outlines a need to address a broader view of the field such as including other osteoclast targeting agents and to outline more therapies that are currently in development that may pave a new path in treating myeloma bone disease. These include anti-sclerostin antibodies and activin A targeting agents that address the need to inform readers of a more modern view and contribute more to the field as noted.

Between lines 271-274, 384-396, and 568-575, we set out to provide such information that addresses developments that align with the future of the field rather than overview less recent developments. Including other potential targets outlined in Bolzoni 2018 PMID: 29495905. These include TRAF6, IL-3, activin A, and BTK which further address the diversity of potential therapies and outline how dynamic this field of research truly is.

As for addressing IMiD and proteosome inhibitors effects on myeloma cells and bone disease, we addressed these omissions in lines 238-245. It became clear how our scope was a bit too focused on bisphosphonate and denosumab therapies as isolated from anti- multiple myeloma therapies that exist which also contribute to improving bone disease in its own right. The two are not mutually exclusive, so a view that acknowledges this fact as a synergy versus discrete regimens is a focus that we believe markedly improves our paper’s scope and effectiveness as an educational tool to inform readers.

Round 2

Reviewer 1 Report

Much improved. However sentence on line 326 is a phrase and not a complete sentence. Appreciate paper added to discuss cost effectiveness of denosumab versus zoledronic acid. 

Author Response

Thank you again for the thoughtful suggested revisions. You comment pertaining to line 326 was addressed by the addition of a conjunction to complete the sentence.

Reviewer 2 Report

The lines regarding proteasome inhibitors are not those reported by the authors (correct lines are 215/222). Moreover, I suggested to mention also inhibitors of sclerostin as therapy targeting bone disease

Author Response

Thank you again for he thoughtful suggested revisions and for noticing the missed line number regarding the proteosome inhibitors.

Lastly, yes I did not correctly address your comment about including sclerostin inhibitors directly. Between lines 399-406, I included a more direct discussion of the sclerostin inhibitor, romosozumab to better outline the trajectory of the field.